# Objective Priors for Invariant *e*-Values in the Presence of Nuisance Parameters

**DOI:** 10.3390/e26010058

**Published:** 2024-01-09

**Authors:** Elena Bortolato, Laura Ventura

**Affiliations:** Department of Statistical Sciences, University of Padova, 35121 Padova, Italy; elena.bortolato@unipd.it

**Keywords:** asymptotic expansions, adjusted score function, bias reduction, evidence, Full Bayesian Significance Test, higher-order asymptotics, matching priors, median bias reduction

## Abstract

This paper aims to contribute to refining the *e*-values for testing precise hypotheses, especially when dealing with nuisance parameters, leveraging the effectiveness of asymptotic expansions of the posterior. The proposed approach offers the advantage of bypassing the need for elicitation of priors and reference functions for the nuisance parameters and the multidimensional integration step. For this purpose, starting from a Laplace approximation, a posterior distribution for the parameter of interest is only considered and then a suitable objective matching prior is introduced, ensuring that the posterior mode aligns with an equivariant frequentist estimator. Consequently, both Highest Probability Density credible sets and the *e*-value remain invariant. Some targeted and challenging examples are discussed.

## 1. Introduction

In this article, we discuss an objective matching prior that maintains the invariance of the posterior mode when testing specific hypotheses for parametric models, especially in the presence of nuisance parameters. These parameters are often introduced to establish flexible and realistic models, although the primary focus of inference is typically limited to a parameter of interest. The proposed approach offers the advantage of eliminating the need for eliciting information on the nuisance components and for conducting multidimensional integration, and it produces invariant e-values in the presence of nuisance parameters.

The parametric framework that we consider can be described as follows. Consider a random sample $y=(y_{1},\ldots ,y_{n})$ of size *n* from a random variable *Y* with parametric model f(y;θ), indexed by a paramater θ, with $\theta \in \Theta \subseteq I\;R^{d}$. Given a prior π(θ) on θ, Bayesian inference for θ is based on the posterior density
(1)$$\begin{array}{c} \pi (\theta |y)\propto \pi (\theta )L(\theta ), \end{array}$$
where L(θ) represents the likelihood function based on f(y;θ). Interest is, in particular, in the situation in which θ=(ψ,λ), where ψ is a scalar parameter for which inference is required, and λ represents the remaining (d−1) nuisance parameters. In such case, Bayesian inference for ψ is based on the marginal posterior density
(2)$$\begin{array}{c} \pi_{m}\left(\psi |y\right)=\int \pi \left(\psi ,\lambda |y\right)\;d\lambda \propto \int \pi \left(\psi ,\lambda \right)L\left(\psi ,\lambda \right)\;d\lambda , \end{array}$$
which for its computation requires both elicitation on the nuisance parameter λ and multidimensional integration.

Asymptotic arguments are widely used in Bayesian inference through (Equation 1) and (Equation 2), based on developments of so-called higher-order asymptotics (see, e.g., [1,2,3]). Indeed, the theory of asymptotic expansions provides very accurate approximations to posterior distributions, and to various summary quantities of interest, including tail areas, credible regions and for the Full Bayesian Significance Test (see, e.g., [4,5]). Moreover, they are particularly useful for sensitivity analyses (see [6,7]) and also for the derivation of matching priors (see [8], and references therein). For instance, focusing on the presence of nuisance parameters, the Laplace approximation to (Equation 2) provides
(3)$$\begin{array}{c} \pi_{m}\left(\psi |y\right)\;\;\ddot{=}\;\;\frac{1}{\sqrt{2\pi }}{\left|j_{p}\left(\hat{\psi }\right)\right|}^{1/2}\exp\left\{{𝓁}_{p}\left(\psi \right)-{𝓁}_{p}\left(\hat{\psi }\right)\right\}\frac{|j_{\lambda \lambda }\left(\hat{\psi },\hat{\lambda }\right){|}^{1/2}}{|j_{\lambda \lambda }\left(\psi ,{\hat{\lambda }}_{\psi }\right){|}^{1/2}}\frac{\pi (\psi ,{\hat{\lambda }}_{\psi })}{\pi (\hat{\psi },\hat{\lambda })}, \end{array}$$
where ${𝓁}_{p}\left(\psi \right)=\log L_{p}\left(\psi \right)=\log L\left(\psi ,{\hat{\lambda }}_{\psi }\right)$ is the profile log-likelihood for ψ, with ${\hat{\lambda }}_{\psi }$ the constrained maximum likelihood estimate (MLE) of λ given ψ, $\left(\hat{\psi },\hat{\lambda }\right)$ is the full MLE, and $j_{p}\left(\psi \right)=-\partial^{2}{𝓁}_{p}\left(\psi \right)/\partial \psi^{2}$ is the profile observed information. Moreover, $j_{\lambda \lambda }\left(\psi ,\lambda \right)$ is the (λ,λ)-block of the observed information from the full log-likelihood 𝓁(ψ,λ)=logL(ψ,λ), and the notation $\ddot{=}$ indicates that the approximation is accurate to order $O(n^{-3/2})$ in moderate deviation regions (see, e.g., [9], Chapter 2). One appealing feature of higher-order approximations like (Equation 3) is that they may routinely be applied in practical Bayesian inference, since they require little more than standard likelihood quantities for their implementation, and hence, they may be available at little additional computational cost over simple first-order approximations.

In the presence of nuisance parameters, starting from approximation (Equation 3), it is possible to define a general posterior distribution for ψ of the form
(4)$$\begin{array}{c} \pi^{*}\left(\psi |y\right)\propto \pi^{*}\left(\psi \right)L_{p}\left(\psi \right), \end{array}$$
where $\pi^{*}\left(\psi \right)$ is now a prior distribution on ψ only. Bayesian inference based on pseudo-likelihood functions—i.e., functions of ψ only and of the data *y* with properties similar to those of a genuine likelihood function, such as the profile likelihood—have been widely used and discussed in the recent statistical literature. Moreover, it has been theoretically motivated in several papers (see, for instance, [10,11,12], and references therein), also focusing on the derivation of suitable objective priors. Especially when the dimension of λ is large, there are two advantages in using (Equation 4) instead of the marginal posterior distribution (Equation 2). First, the elicitation over λ is not necessary, and second, the computation of the integrals in (Equation 2) is circumvented.

Focusing on (Equation 4), in this paper, it is of interest to test the precise (or sharp) null hypothesis
(5)$$\begin{array}{c} H_{0}:\psi =\psi_{0}\;\mathrm{against}\;H_{1}:\psi \neq \psi_{0} \end{array}$$
using the measure of evidence for the Full Bayesian Significance Test (see, e.g., [4,5]). The Full Bayesian Significance Test (FBST) quantifies evidence by considering the posterior probability associated with the least probable points in the parameter space under $H_{0}$. Higher-order asymptotic computation of the FBST for precise null hypotheses in the presence of nuisance parameters has been discussed in [13].

The original measure of evidence for the FBST is not invariant under suitable transformations of the parameter, a property which has, however, been reached in the more recent definition of the e-value (see [14,15], and references therein). Neverthless, when working on a scalar parameter of interest, in the presence of nuisance parameters, the e-value is not invariant with respect to marginalizations of the nuisance parameter, and it must be used in the full dimensionality of the parameter space. This requires elicitation on the complete parameters, numerical optimization and numerical integration, that can be computationally heavy, especially when the dimension of λ is large.

The aim of this paper is to consider the e-value in the context of the pseudo-posterior distribution $\pi^{*}\left(\psi |y\right)$, suggesting in this respect a suitable objective prior $\pi^{*}\left(\psi \right)$ to be used in (Equation 4). More precisely, focus is on a particular matching prior, which ensures the invariance of the posterior mode of the pseudo-posterior distribution. As a consequence, Highest Probability Density credible (HPD) sets are also invariant, as well as the e-value.

This paper is organized as follows. Section 2 provides a short review on the FBST for testing precise null hypotheses and also illustrates asymptotic approximations for the e-value, extending results of [13]. Section 3 discusses the derivation of the objective matching prior for the parameter of interest only, called median matching prior, that produces invariant e-values. Also, several targeted and challenging examples are discussed. Finally, Section 4 closes with some concluding remarks.

## 2. The FBST Measure of Evidence

Suppose that we need to decide between two hypotheses: the null $H_{0}$ and the alternative $H_{1}$. The usual Bayesian testing procedure is based on the well-known Bayes factor (BF), defined as the ratio of the posterior odds to the prior odds in favor of the null hypothesis. A high BF or its logarithm suggests evidence in favor of $H_{0}$. However, it is well known that, when improper priors are used, the BF can be undetermined, and when the null hypothesis is precise (as specified in (Equation 5)), the BF can lead to the so-called Jeffreys–Lindley’s paradox (see, e.g., [16]). Moreover, the BF is not calibrated, i.e., its finite sampling distribution is unknown and it may depend on the nuisance parameter.

To avoid these drawbacks, in recent years, an alternative Bayesian procedure, called FBST, has been introduced by [5] in case of sharp hypothesis $H_{0}$ identified by the null set $\Theta_{0}$, a submanifold of Θ of lower dimension. The FBST quantifies evidence by considering the posterior probability associated with the least probable points in the parameter space $\Theta_{0}$. When this probability is high, it favors the null hypothesis, providing a clear and interpretable measure of support for $H_{0}$ (see, e.g., [4,15,17], and references therein). The FBST is based on a specific loss function, and thus, the decision made under this procedure is the action that minimizes the corresponding posterior risk.

The FBST operates by determining the e-value, a representation of Bayesian evidence associated to $H_{0}$. To construct the e-value, the authors introduced the *posterior surprise function* and its supremum given, respectively, by
$$\pi_{s}\left(\theta |y\right)=\frac{\pi (\theta |y)}{r(\theta )}\;\mathrm{and}\;s^{*}=\pi_{s}\left(\theta^{*}|y\right)=\underset{\theta \in \Theta_{0}}{\sup}\pi_{s}\left(\theta |y\right),$$
where r(θ) is a suitable *reference function* to be chosen. The surprise function was introduced in the context of statistical inference also by [18] (see [15], and references therein). Then, they introduce the *tangential set* $T_{y}\left(\theta^{*}\right)$ defined as the set of parameter values for which the posterior surprise function exceeds the supremum $s^{*}$, that is
$$T_{y}\left(\theta^{*}\right)=\left\{\theta \in \Theta :\pi_{s}\left(\theta |y\right)>s^{*}\right\}.$$
This set, often referred to as the Highest Relative Surprise Set, includes parameter values with higher surprise than those within the null set $\Theta_{0}$. The e-value is then computed as
$$ev=1-\int_{T_{y}\left(\theta^{*}\right)}\pi_{s}\left(\theta |y\right)\;d\theta ,$$
and $H_{0}$ is rejected for *small* values of ev.

The original FBST, as proposed by [5,19], relies on a flat reference function r(θ)∝1, so that this first version involved the determination of the tangential set $T_{y}\left(\theta \right)$ starting only from the posterior distribution π(θ|y). However, this initial version lacked invariance under reparameterizations. Subsequent refinements of the FBST introduced the importance of reference density functions, making the e-value explicitly invariant under appropriate transformations of the parameter. Common choices for the reference function include uninformative priors, like the uniform distribution, maximum entropy densities, or Jeffreys’ invariant prior. In [20], the use of the Jeffreys’ prior, $\pi \left(\theta \right)\propto {\left|i\left(\theta \right)\right|}^{1/2}$, where i(θ) is the Fisher information derived from L(θ), is discussed as the reference function to derive invariant HPD sets and Maximum A Posteriori (MAP) estimators that are invariant under reparameterizations. Note that the ev uses the full dimensionality of the parameter space. Moreover, this measure is not invariant with respect to transformations of the nuisance parameters, and the use of high posterior densities to construct credible sets may produce inconsistencies.

Concerning the asymptotic behavior of the ev, it can be proven that, under suitable regularity conditions as the sample size increases, with $\theta_{0}$ representing the true parameter value (see [15]), it holds:If $H_{0}$ is false, i.e., $\theta_{0}\notin H_{0}$, then ev converges in probability to 1.If $H_{0}$ is true, i.e., $\theta_{0}\in H_{0}$, then, denoting by V(c)=Pr(ev≤c) the cumulative distribution function of ev, we have that $V\left(c\right)\approx Q(d-h,Q^{-1}\left(d,c\right))$, with d=dim(Θ), $h=dim(\Theta_{0})$ and Q(k,x) the cumulative chi-square distribution with *k* degrees of freedom.

In practice, the computation of ev is performed in two steps: (a) a numerical optimization and (b) a numerical integration. The numerical optimization step consists of finding the maximizer $\theta^{*}$ of $\pi_{s}\left(\theta |y\right)$ under the null hypothesis. The numerical integration step consists of integrating the posterior surprise function over the region where it is greater than $\pi_{s}\left(\theta^{*}|y\right)$, to obtain the e-value. These computational steps make the FBST a computationally intensive procedure. Despite efficient computational algorithms for local and global optimization, as well as numerical integration, obtaining precise results for hypotheses like (Equation 5) is highly demanding, especially with large nuisance parameter dimensions. Numerical integration can be tackled by resorting to higher-order tail area approximations, as reviewed in the Bayesian framework in [3]. An application of asymptotic approximation to the FBST in its first formulation, i.e., with reference function r(θ)∝1, has been discussed in [13].

### Asymptotic Approximations for the e-Value

A first-order approximation for the e-value, when testing (Equation 5), is simply given by (see, e.g., [21,22])
(6)$$\begin{array}{c} ev\;\;\dot{=}\;\;2\left(1-\Phi \left(\left|\frac{\psi_{0}-\hat{\psi }}{\sqrt{j_{p}{\left(\hat{\psi }\right)}^{-1}}}\right|\right)\right), \end{array}$$
where the symbol “$\dot{=}$” indicates that the approximation is accurate to $O(n^{-1/2})$, and Φ(·) is the standard normal distribution function. Thus, to first-order, ev agrees with the p-value based on the profile Wald statistic
(7)$$\begin{array}{c} w_{p}\left(\psi \right)=\frac{\left(\hat{\psi }-\psi_{0}\right)}{\sqrt{j_{p}{\left(\hat{\psi }\right)}^{-1}}}. \end{array}$$
In practice, the approximation (Equation 6) of ev may be inaccurate, in particular when the dimension of λ is large with respect to the sample size, because it forces the marginal posterior distribution to be symmetric.

The practical computation of ev requires the evaluation of integrals of the marginal posterior distribution. In order to have more accurate evaluations of ev, it may be useful to resort to higher-order asymptotics based on tail area approximations (see, e.g., [2,3], and references therein). Indeed, the measure of evidence involves integrals of the marginal surprise posterior density $\pi_{ms}\left(\psi |y\right)$. In particular, extending the application of the tail area argument to the marginal surprise posterior density, we can derive a $O(n^{-3/2})$ approximation to the marginal surprise posterior tail area probability, given by
(8)$$\begin{array}{c} \int_{\psi_{0}}^{\infty }\pi_{ms}\left(\psi |y\right)\;d\psi \;\;\ddot{=}\;\;\Phi \left(r_{B}^{*}\left(\psi_{0}\right)\right), \end{array}$$
where
$$r_{B}^{*}\left(\psi \right)=r_{p}\left(\psi \right)+\frac{1}{r_{p}\left(\psi \right)}\log\frac{q_{B}\left(\psi \right)}{r_{p}\left(\psi \right)},$$
with
$$r_{p}\left(\psi \right)=\mathrm{sign}\left(\hat{\psi }-\psi \right){\left[2\left({𝓁}_{p}\left(\hat{\psi }\right)-{𝓁}_{p}\left(\psi \right)\right)\right]}^{1/2}$$
profile likelihood root and
$$q_{B}\left(\psi \right)={𝓁}_{p}^{\prime }\left(\psi \right){\left|j_{p}\left(\hat{\psi }\right)\right|}^{-1/2}\frac{|j_{\lambda \lambda }\left(\psi ,{\hat{\lambda }}_{\psi }\right){|}^{1/2}}{|j_{\lambda \lambda }\left(\hat{\psi },\hat{\lambda }\right){|}^{1/2}}\frac{\pi (\hat{\psi },\hat{\lambda })}{\pi (\psi ,{\hat{\lambda }}_{\psi })}\frac{r(\psi ,{\hat{\lambda }}_{\psi })}{r(\hat{\psi },\hat{\lambda })}.$$
In the expression of $q_{B}\left(\psi \right)$, ${𝓁}_{p}^{\prime }\left(\psi \right)=\partial {𝓁}_{p}\left(\psi \right)/\partial \psi$ is the profile score function.

Using the tail area approximation (Equation 8), a third-order approximation of the measure of evidence ev can be derived. The approximation, assuming without loss of generality that $\psi_{0}$ is smaller than the MAP of $\pi_{ms}\left(\psi |y\right)$, is given by
(9)$$\begin{array}{c} ev\left(\psi \right)\;\;\ddot{=}\;\;1-\Phi \left(r_{B}^{*}\left(\psi_{0}\right)\right)+\Phi \left(r_{B}^{*}\left(\psi_{0}^{*}\right)\right), \end{array}$$
with $\psi_{0}^{*}$ the value of the parameter such that $\pi_{ms}\left(\psi_{0}^{*}|y\right)=\pi_{ms}\left(\psi_{0}|y\right)$. Note that
$$\Phi \left(r_{B}^{*}\left(\psi_{0}\right)\right)-\Phi \left(r_{B}^{*}\left(\psi_{0}^{*}\right)\right)\;\ddot{=}\;\int_{\psi_{0}^{*}}^{\psi_{0}}\pi_{ms}\left(\psi |y\right)\;d\psi =1-ev$$
in (Equation 9) gives the posterior probability of the HPD credible interval $\left(\psi_{0},\psi_{0}^{*}\right)$. Note also that the higher-order approximation (Equation 9) does not call for any condition on the prior π(ψ,λ), i.e., it can be also improper. Finally, when $\pi_{ms}\left(\psi |y\right)$ is symmetric, Equation (Equation 9) reduces to $ev\;\ddot{=}\;2\left(1-\Phi \left(r_{B}^{*}\left(\psi_{0}\right)\right)\right)$.

While tail area approximations require little more than standard likelihood quantities for their implementation and, in this respect, they are available at little additional computational cost over the first-order approximation, they require elicitation on the complete parameter θ and to choose the reference function r(θ).

## 3. An Invariant Objective Prior

The aim of this section is to derive a default prior $\pi^{*}\left(\psi \right)$ to be used in (Equation 4). To this end, following [8], we use the shrinkage argument, which is a crucial procedure in the development of matching priors, i.e., priors that ensure, up to the desired order of asymptotics, an agreement between Bayesian and frequentist procedures. Examples of matching priors are (see [8]) for posterior quantiles, for credible regions and for prediction. Here, we focus on a specific matching prior that ensures the invariance of the posterior mode in the posterior distribution (Equation 4). As a consequence, the invariance extends to HPDs, as well as the e-value, achieved incorporating the reference function within the prior.

The proposed choice of the prior $\pi^{*}\left(\psi \right)$, which makes the MAP and thus also HPDs and the e-value invariant under 1-1 reparameterization, will depend on the log-likelihood 𝓁(θ) and on its derivatives. In regular parametric estimation problems, both the MLE and the score-estimating function exhibit an asymptotically symmetric distribution centered at the true parameter value and at zero, respectively. However, these asymptotic behaviors may poorly reflect exact sampling distributions, particularly in cases with small or moderate sample information, sparse data, or complex models. Several proposals have been developed to correct the estimate or the estimating function. Most available methods are aimed at approximate bias adjustment, either of the MLE or of the profile score function, also when nuisance parameters are present (see [23] for a review of bias reduction for the MLE and [24] and subsequent literature for bias correction of the profile score). Lack of equivariance impacts the so-called implicit bias reduction methods, which achieve first-order bias correction by modifying the score equation (see [23,25]). To avoid this drawback, in this paper we focus on the median modification of the score, or profile score equation, whose solution respects equivariance under monotone reparameterizations ([26]). Similar to Firth’s implicit method ([25]), the median modification of the score, or profile score, does not rely on finiteness of the MLE, thereby effectively preventing infinite estimates.

In practice, to derive the median matching prior $\pi^{*}\left(\psi \right)$, we impose that the MAP of $\pi^{*}\left(\psi |y\right)$ coincides with a refined version of the MLE, obtained as the solution of the median modified score function ([26]). To introduce this new invariant prior, we initially explore the scenario without nuisance parameters and then the situation in which nuisance parameters are present.

### 3.1. No Nuisance Parameters

Let us explore first the scenario where θ is scalar. In order to obtain median bias reduction of the MLE, it is possible to resort to a modified version of the score function of the form
(10)$$\begin{array}{c} t\left(\theta \right)={𝓁}_{\theta }\left(\theta \right)+m\left(\theta \right), \end{array}$$
where ${𝓁}_{\theta }\left(\theta \right)={𝓁}_{\theta }\left(\theta ;y\right)=\partial 𝓁\left(\theta ;y\right)/\partial \theta$ is the score function, and m(θ) is a suitable correction term of order O(1). In particular, the median modified score function assumes for m(θ) the expression
$$m\left(\theta \right)=-\frac{E({𝓁}_{\theta }{\left(\theta \right)}^{3})}{6\;i(\theta )}.$$
The solution $\tilde{\theta }$ to the equation t(θ)=0 not only upholds equivariance under componentwise monotone reparameterizations, but also approximates median unbiasedness ([26]). Note that likelihood inference based on (Equation 10) does not depend explicitely on the MLE. Indeed, the modified score function has been found to overcome infinite estimate problems. Likewise, the MLE and also $\tilde{\theta }$ is asymptotically $N(\theta ,i{\left(\theta \right)}^{-1})$, so that the Wald-type statistics only differ in location.

Since Bayes’ theorem is a statement of adittivity on the log scale logπ(θ|y)=logπ(θ)+logL(θ)+ constant, we observe that in the Bayesian framework, m(θ) can be interpreted as the derivative of the logarithm of a prior, that is, m(θ)=∂logπ(θ)/∂θ. We are thus looking for a matching prior $\pi^{*}\left(\theta \right)$ such that
$$\frac{\partial \log\pi^{*}\left(\theta \right)}{\partial \theta }=-\frac{E({𝓁}_{\theta }{\left(\theta \right)}^{3})}{6\;i(\theta )}.$$
In the scalar parameter case, it is straightforward to show that the proposed *median matching prior* takes the form
$$\begin{array}{ccc} \pi^{*}\left(\theta \right) & \propto & \exp\left(-\frac{1}{6}\int i{\left(\theta \right)}^{-1}E\left({𝓁}_{\theta }{\left(\theta \right)}^{3}\right)\;d\theta \right)\\ & \propto & \exp\left(\frac{1}{6}\int i{\left(\theta \right)}^{-1}\left(3E\left({𝓁}_{\theta \theta }\left(\theta \right){𝓁}_{\theta }\left(\theta \right)\right)+E\left({𝓁}_{\theta \theta \theta }\left(\theta \right)\right)\right)\;d\theta \right), \end{array}$$
with ${𝓁}_{\theta \theta }\left(\theta \right)=\partial {𝓁}_{\theta }\left(\theta \right)/\partial \theta$ and ${𝓁}_{\theta \theta \theta }\left(\theta \right)=\partial {𝓁}_{\theta \theta }\left(\theta \right)/\partial \theta$, where the second expression for $\pi^{*}\left(\theta \right)$ follows from the Bartlett’s identities. The posterior based on the median matching prior is thus
$$\pi^{*}\left(\theta |y\right)\propto \exp\left(𝓁\left(\theta \right)-\frac{1}{6}\int i{\left(\theta \right)}^{-1}E\left({𝓁}_{\theta }{\left(\theta \right)}^{3}\right)\;d\theta \right).$$

A first-order approximation for the e-value, when testing $H_{0}:\theta =\theta_{0}$, is simply given by
(11)$$\begin{array}{c} ev\;\;\dot{=}\;\;2\left(1-\Phi \left(\left|\frac{\theta_{0}-\tilde{\theta }}{\sqrt{i{\left(\theta_{0}\right)}^{-1}}}\right|\right)\right), \end{array}$$
which differs in location with respect to the classical first-order approximation for the e-value based on the MLE. A second approximation for the e-value, when testing $H_{0}:\theta =\theta_{0}$, can be obtained from the asymptotic distribution of the modified score function (Equation 10), that is
(12)$$\begin{array}{c} ev\;\;\dot{=}\;\;2\left(1-\Phi \left(\left|\frac{t(\theta_{0})}{\sqrt{i(\theta_{0})}}\right|\right)\right). \end{array}$$
Although the first-order equivalence between (Equation 11) and (Equation 12), note that (Equation 11) is based on an easily understandable comparison between estimated value and hypothetical value, taking estimation error into account, and is widely used in applications but does not satisfy the principle of parameterization invariance. On the other hand, $t\left(\theta \right)/\sqrt{i(\theta )}$ is parameterization invariant.

Note that, when using a *predictive matching prior*, i.e., a prior ensuring asymptotic equivalence of higher-order frequentist and Bayesian predictive densities (see, e.g., [8]), the term m(θ) in (Equation 10) corresponds to the Firth’s adjustment ([25])
$$m_{F}\left(\theta \right)=-\frac{\left(E\left({𝓁}_{\theta }{\left(\theta \right)}^{3}\right)+E\left({𝓁}_{\theta \theta }\left(\theta \right){𝓁}_{\theta }\left(\theta \right)\right)\right)}{2i(\theta )}.$$
In view of this, for general regular models, Firth’s estimate coincides with the mode of the posterior distribution obtained using the default predictive matching prior. However, lack of invariance affects this kind of adjustment ([23]), unless dealing with linear transformations.

**Example** **1**
**(One parameter exponential family).**
* For a one-parameter exponential family with canonical parameter θ, i.e., with density*

f(y;θ)=exp{θa(y)−K(θ)}b(y),

*the median modified score function has the form*

$$t\left(\theta \right)={𝓁}_{\theta }\left(\theta \right)+\frac{K_{\theta \theta \theta }}{6K_{\theta \theta }},$$

*where $K_{\theta \theta \theta }=\partial^{3}K\left(\theta \right)/\partial \theta^{3}$ and $K_{\theta \theta }=\partial^{2}K\left(\theta \right)/\partial \theta^{2}=i\left(\theta \right)$. In this parameterization, t(θ) can be seen as the first derivative of the log-posterior*

logπ(θ|y)=𝓁(θ)+logi(θ)/6.

*On the other hand, Firth’s modified score takes the form $t_{F}\left(\theta \right)={𝓁}_{\theta }\left(\theta \right)+K_{\theta \theta \theta }/\left(2K_{\theta \theta }\right)$. The effect of the median modification is to consider the median matching prior $\pi^{*}\left(\theta \right)\propto i{\left(\theta \right)}^{1/6}$, while $t_{F}\left(\theta \right)$ implies a Jeffreys’ prior $\pi_{J}\left(\theta \right)\propto i{\left(\theta \right)}^{1/2}$. Note that, for a one-parameter exponential family with canonical parameter θ, both $\pi^{*}\left(\theta \right)$ and $\pi_{J}\left(\theta \right)$ belong to the family of invariant priors discussed in [27,28].*


**Example** **2**
**(Scale model).**
* Consider the scale model $f\left(y;\theta \right)=\left(1/\theta \right)p_{0}\left(y/\theta \right)$, where $p_{0}\left(\cdot \right)$ is a given function. Let $g\left(\cdot \right)=-\log p_{0}\left(\cdot \right)$. We have $E\left({𝓁}_{\theta }^{3}\right)=c_{1}/\theta^{3}$, $E\left({𝓁}_{\theta }^{3}{𝓁}_{\theta }\right)=c_{2}/\theta^{3}$ and $i\left(\theta \right)=c_{3}/\theta^{2}$, with $c_{1}=\int \left(y^{3}g^{\prime \prime \prime }\left(y\right)+6y^{2}g^{\prime \prime }\left(y\right)+6yg^{\prime }\left(y\right)-2\right)p_{0}\left(y\right)dy$, $c_{2}=\int \left(3yg^{\prime }\left(y\right)+y^{2}g^{\prime \prime }\left(y\right)-2y^{2}g^{\prime }{\left(y\right)}^{2}-y^{3}g^{\prime }\left(y\right)g^{\prime \prime }\left(y\right)-1\right)p_{0}\left(y\right)dy$ and $c_{3}=\int \left(2yg^{\prime }\left(y\right)+y^{2}g^{\prime \prime }\left(y\right)-1\right)p_{0}\left(y\right)dy$. The median matching prior is thus $\pi^{*}\left(\theta \right)\propto \theta^{-c_{1}/6c_{3}}$, while the Jeffreys’ prior for a one-parameter scale model is $\pi_{J}\left(\theta \right)\propto \theta^{-1}$ and the prior associated to the Firth’s adjustment is $\pi_{F}\left(\theta \right)\propto \theta^{-\left(c_{1}+c_{2}\right)/2c_{3}}$.*


**Example** **3**
**(Skew–normal distribution).**
* Consider a skew-normal distribution with shape parameter θ∈IR, with density f(y;θ)=2ϕ(y)Φ(yθ), where ϕ(·) is the standard normal density function. The median correction term for the score function associated to the median matching prior is (see [26,29])*

$$m\left(\theta \right)=\frac{E(y^{3}\phi {\left(y\theta \right)}^{3}/\Phi {\left(y\theta \right)}^{3})}{6E(y^{2}\phi {\left(y\theta \right)}^{2}/\Phi {\left(y\theta \right)}^{2})}.$$

*Numerical integration must be performed to obtain the expected values involved in m(θ).*

*In order to illustrate the proposed prior, we consider draws from the skew–normal model with true parameter $\theta_{0}=3$ and increasing sample sizes n=20,30,50,200 (Figure 1). The posterior distributions are obtained with the method by [30], i.e., drawing $10^{5}$ values and accepting the best 5%. The e-values associated to the null (true) hypothesis $H_{0}:\theta =3$ and the (false) hypothesis $H_{0}:\theta =4$ are reported in Table 1. For comparison, the Jeffreys’ prior ([31]), the predictive matching prior ([29]) and the flat prior, with uniform reference function, are also considered. Progressive agreement among evidence values obtained with the proposed median matching prior and the other priors for larger sample size is shown. Also, as expected, when progressively increasing n, the evidence values indicate agreement with the true hypothesis $H_{0}:\theta_{0}=3$ and disagreement with $H_{0}:\theta_{0}=4$ for all the priors used. Anyway, note that the posterior distribution obtained with a flat prior and a uniform reference function is proportional to the likelihood function that can be monotone. In view of this, while the MAPs of the posterior based on the default priors are always finite, in some samples the MAP of the posterior with the non-informative prior may be infinite. An example of this effect is illustrated in Figure 2.*

*The properties of first-order approximations of the e-values have been investigated by a simulation study, with sample sizes n=20,30,50,200. Results are displayed in Figure 3. Distributions of the e-value from the posterior based on the median matching prior are better, both for small and moderate sample sizes, in terms of convergence to the uniform distribution. Moreover, score-type e-values (Equation 12) are also preferable over Wald-type e-values (Equation 11). For the results with the posterior distribution obtained with a flat prior, we found 4.3%,4.2%,0.9%,0% of infinite estimates for the sample sizes considered in the simulation study, and in these cases the e-value was considered as 0.*


### 3.2. Presence of Nuisance Parameters

In the presence of nuisance parameters, in order to obtain median bias reduction of the MLE, it is possible to resort to a modified version of the profile score function of the form
(13)$$\begin{array}{c} t_{p}\left(\psi \right)={𝓁}_{p}^{\prime }\left(\psi \right)+m\left(\psi ,{\hat{\lambda }}_{\psi }\right), \end{array}$$
where m(ψ,λ) is a suitable correction term of order O(1). In particular, for the median modified profile score function, the adjustment m(ψ,λ) assumes the expression
$$m\left(\psi ,\lambda \right)=-\kappa_{1\psi }+\frac{\kappa_{3\psi }}{6\kappa_{2\psi }},$$
where $\kappa_{1\psi }$, $\kappa_{2\psi }$ and $\kappa_{3\psi }$ are the first three cumulants of ${𝓁}_{p}^{\prime }\left(\psi \right)$ (see [26], Section 2.2, for their expression). For the estimator ${\tilde{\psi }}_{p}$, defined as the solution of $t_{p}\left(\psi \right)=0$, parameterization equivariance holds under interest respecting reparameterizations ([26]).

Note that, also in the context of nuisance parameters, we are in the situation in which the proposed prior $\pi^{*}\left(\psi \right)$ is known through its first derivative; this is typically the situation with matching priors (see, e.g., [8]). Since the parameter of interest is scalar, the posterior based on the median matching prior can be written as
(14)$$\begin{array}{c} \pi^{*}\left(\psi |y\right)\propto \exp\left({𝓁}_{p}\left(\psi \right)+\int m\left(\psi ,{\hat{\lambda }}_{\psi }\right)\;d\psi \right). \end{array}$$

A simple analytical way of approximating to first-order the posterior distribution (Equation 14) based on the median matching prior is to resort to a quadratic form of $t_{p}\left(\psi \right)$. In particular, the approximate posterior distribution for ψ takes the form
(15)$$\begin{array}{c} \pi^{*}\left(\psi |y\right)\;\dot{\propto }\;\exp\left(-\frac{1}{2}s_{p}\left(\psi ;y\right)\right), \end{array}$$
where $s_{p}\left(\psi \right)=t_{p}{\left(\psi \right)}^{2}j_{p}{\left(\psi \right)}^{-1}$ is a Rao score-type statistic based on (Equation 13), and the symbol “$\dot{\propto }$” means asymptotic proportionality to first-order. In this case, a first-order approximation of the e-value, when testing $H_{0}:\psi =\psi_{0}$, is given by
(16)$$\begin{array}{c} ev\;\;\dot{=}\;\;2\left(1-\Phi \left(\left|\frac{t_{p}\left(\psi_{0}\right)}{\sqrt{j_{p}\left(\psi_{0}\right)}}\right|\right)\right). \end{array}$$
In this case, an higher-order approximation via (Equation 3) would be impractical since a closed-form prior is not available. As an alternative, simulation-based approaches may be used to derive the implied posterior distribution (Equation 14) based on the median matching prior. The first one relies on Approximate Bayesian Computation (ABC) techniques, using ${\tilde{\psi }}_{p}$ or the modified profile score function $t_{p}\left(\psi \right)$ as summary statistics; see [32] for the modification of the algorithm of [30] by using a profile estimating equation. This first method introduces an approximation at the level of the posterior estimation. The second one still relies on (Equation 13) but considers use of Manifold MCMC methods (see, e.g., [33]) to conditioning exactly on the profile equation and not up to a tolerance level, as in ABC (see [34,35]). The algorithm moves on the constrained space $\left\{\left(y,\psi \right)\in Y\times ⊖|t_{p}\left({\tilde{\psi }}_{p}\right)=0\right\}$, where ${\tilde{\psi }}_{p}$ is the solution of the estimating equation on the original data. For the latter method, we need minimal regularity assumptions on m(ψ,λ), which is assumed to be continuous, differentiable and available in closed form expression. Note, for instance, that in the skew-normal example in Section 3.1 these conditions are not met.

**Example** **4**
**(Exponential family).**
* If f(y;θ) is an exponential family of order d with canonical parameter (ψ,λ), i.e., $f\left(y;\psi ,\lambda \right)=\exp\{\psi t\left(y\right)+\lambda^{T}s\left(y\right)-K\left(\psi ,\lambda \right)\}h\left(y\right)$, quantities involved in m(ψ,λ) are simply obtained from derivatives of K(ψ,λ) ([26]). Note that, in this framework, ${𝓁}_{p}^{\prime }\left(\psi \right)-\kappa_{1\psi }$ is an approximation with error of order $O(n^{-1})$ of the score for ψ in the conditional model given s(y) (see e.g., [36], Section 10.2). Then, in the continuous case, the MAP ${\tilde{\psi }}_{p}$ is an approximation of the optimal conditional median unbiased estimator, and $\pi^{*}\left(\psi |y\right)$ is related to the conditional likelihood for ψ given by $L_{c}\left(\psi \right)=\exp\left(\psi t\left(y\right)-K_{s}\left(\psi \right)\right)$; see [37] for a Bayesian interpretation of such pseudo-likelihoods.*


**Example** **5**
**(Multivariate regression model).**
* Consider a regression model of the form*

$$Y_{ij}=\beta_{0}+\beta_{1}x_{i1}+\beta_{2}x_{i2}+\epsilon_{ij},\;i=1,\ldots ,n,\;j=1,2,$$

*where it is assumed that $\epsilon_{i}∼N_{2}\left(0,\Sigma \right)$, with $\Sigma =\sigma^{2}\left(\begin{array}{cc} 1 & \rho \\ \rho & 1 \end{array}\right)$ positive definite matrix, and $\left(\beta_{0},\beta_{1},\beta_{2},\sigma^{2},\rho \right)$ are unknown parameters. This model is widely used for instance in time series analysis, in which as regression covariates the past of the observables y are used. We focus on the problem of testing hypothesis on the correlation coefficient ρ.*

*Consider a draw with true parameter $\rho_{0}=0.95$ and n=20. For obtaining the proposed posterior (Equation 14) for ρ, we first compute the MAPs with the median matching prior and also, for comparison, with the predictive matching prior, which are, respectively, 0.953 and 0.92. Note that the expression of the predictive matching prior for (ψ,λ) corresponds to the Firth’s adjustment to the score function. The expressions of the modified profile estimating functions $t_{p}\left(\psi \right)$ and $t_{F}\left(\psi \right)$ are obtained from [38] and are available in closed form expressions. Hence, the Manifold MCMC method can be used to obtain the implied posteriors, whose approximation is comparable to that of any MCMC sampler. In particular we used 20,000 iterations.*
*We compare the posterior distribution based on the proposed median matching prior, with those obtained with the predictive matching prior and with an inverse-Wishart prior for the covariance matrix* Σ *with one degree of freedom and identity position, and uniform prior on the regression parameters. The posterior distributions are displayed in Figure 4. The hypothesis of interest is $H_{0}:\rho =0.9$, and a smaller e-value indicating disagreement with the hypothesis should be preferable. The e-values are 0.25 with the median matching prior, 0.36 for the predictive matching prior and 0.60 with the inverse-Wishart prior. Note that the e-value based on the inverse-Wishart prior involves the constrained maximization and multidimensional integration and thus is not directly readable in Figure 4. Indeed, one crucial difference is that the original e-value formulation links the evidence of the null hypothesis to the evidence of a more refined hypothesis, choosing the MAP under the null hypothesis for all the nuisance parameters, while in the alternative (tangential) set, all values are used, and integration is performed on the full dimensionality of the space. On the contrary, in the proposed posterior based on the median matching prior, the maximizer of nuisance parameters are taken both in the null and non-null sets.*
*Finally, for the posterior based on the inverse-Wishart prior, we also computed the e-value based on the high-order tail area approximation (Equation 9) of the marginal surprise posterior, which is equal to 0.27. This procedure still avoids the multidimensional integration, but the result is not invariant to changes in parametrization.*


**Example** **6**
**(Logistic regression model).**
* Let $y_{i}$, i=1,…,n, be independent realizations of binary random variables with probability $\pi_{i}$, where $\log\left(\pi_{i}/\left(1-\pi_{i}\right)\right)=\eta_{i}=x_{i}\beta$ and $x_{i}=\left(x_{i1},\ldots ,x_{ip}\right)$ is a row vector of covariates. We assume that a generic scalar component of β is of interest, and we treat the remaining components as nuisance parameters.*
*As an example, we consider the* endometrial cancer grade *dataset analyzed, among others, in [39]. The aim of the clinical study was to evaluate the relationship between the histology of the endometrium (HG), the binary response variable, of n=79 patients and three risk factors: 1. Neovascularization (NV), that indicates the presence or extent of new blood vessel formation; 2. Pulsatility Index (PI), that measures blood flow resistance in the endometrium; 3. Endometrium Height (EH), that indicates the thickness or height of the endometrium. A logistic model for HG, including an intercept and using all the covariates (NV, PI, EH), has been fitted, but a maximum likelihood leads to the infinite MLE of the coefficient $\beta_{2}$ related to NV, due to quasi-complete separation. This phenomenon prohibits the use of diffuse priors for $\beta_{2}$, since the corresponding posterior would not concentrate. Moreover, the e-value with non-informative priors cannot be obtained also for any hypothesis concerning parameters different from $\beta_{2}$.*
*If we consider $\beta_{2}$ as the parameter of interest, while the remaining regression coefficients are treated as nuisance parameters, the analysis with the median matching prior allows us to obtain a global proper posterior, with MAP equal to 3.86, open to interpretation both in the original scale and in terms of odds ratios. Similarly, the posterior based on the predictive matching prior, which in this model coincides with Jeffreys’ prior $\pi \left(\beta \right)\propto {\left|i\left(\beta \right)\right|}^{1/2}$ is proper, with the MAP set at 2.92. The latter suffers from a lack of interpretability on different scales, since a different parametrization in the estimation phase would affect the results.*

*If we consider $\beta_{3}$ as the parameter of interest, related to the risk factor PI, the MAPs are −0.038 when using the median matching prior and −0.035 when using the predictive matching prior. The e-values for the hypothesis $H_{0}:\beta_{3}=0$ are 0.60 and 0.55, respectively (see Figure 5). Likewise, the interpretation of e-values remains consistent and independent of parametrization solely in the first case.*


## 4. Conclusions

Although (Equation 14) cannot always be considered as orthodox in a Bayesian setting, the use of alternative likelihoods is nowadays widely shared, and several papers focus on the Bayesian application of some well-known pseudo-likelihoods. In particular, the proposed posterior $\pi^{*}\left(\psi |y\right)$ has the advantages of avoiding the elicitation on the nuisance parameter λ and of the computation of multidimensional integrations. Moreover, it provides invariant MAPs, HPDs and e-values, without the adoption of a reference function. Finally, we remark that frequentist properties of the MAP of the posterior based off the proposed median matching prior in comparison with the MAP of the posterior based of the predictive matching prior have been investigated in [26,38] for some of the examples discussed in this paper.

For inference on a full vector parameter θ, with d>1 components, a direct extension of the rationale leading to (Equation 10) does not seem to be practicable due to lack of a manageable definition of the multivariate median. Actually, in [26,40], it is shown how the method can be extended to a vector parameter of interest in the presence of nuisance parameters by simultaneously solving median bias-corrected score equations for all parameter components. This leads to componentwise third-order median unbiasedness and parameterization equivariance. Moreover, the use of default priors involving all parameter components, also the nuisance, becomes necessary to regularize likelihoods in case of monotonicity. We note that among the possible objective priors that ensures invariance of the posterior, we did not focus on the Jeffreys’ in the multidimensional case, since it often exhibits poor convergence properties. Conversely, the default matching priors considered in this paper are easily generalizable to the multidimensional case [40] preserving good convergence properties.

As a final remark, we highlight that this paper opens several topics of future research. In particular, from a computational point of view, it could be of interest:To develop a library of computational routines exploring the methods proposed in this paper for a wide range of statistical models of interest;To develop semi-automated procedures for further expanding this library, as is done for point estimation for Generalized Linear Models in the R package brglm2 [41].
Moreover, from a theoretical point of view, it could be of interest:To further explore the theoretical connections between the e-value invariance properties and matching priors;To explore the existence of similar connections in other classes of pseudo-likelihoods, in particular in the context of empirical and profile empirical likelihoods, with a large number of nuisance parameters ([42]),To apply and extend the methodology to consider other objective priors used in Bayesian inference, such as those obtained from scoring rules, as proposed in [10], that are expressed as solutions of differential equations.

## Figures and Tables

**Figure 1 entropy-26-00058-f001:**
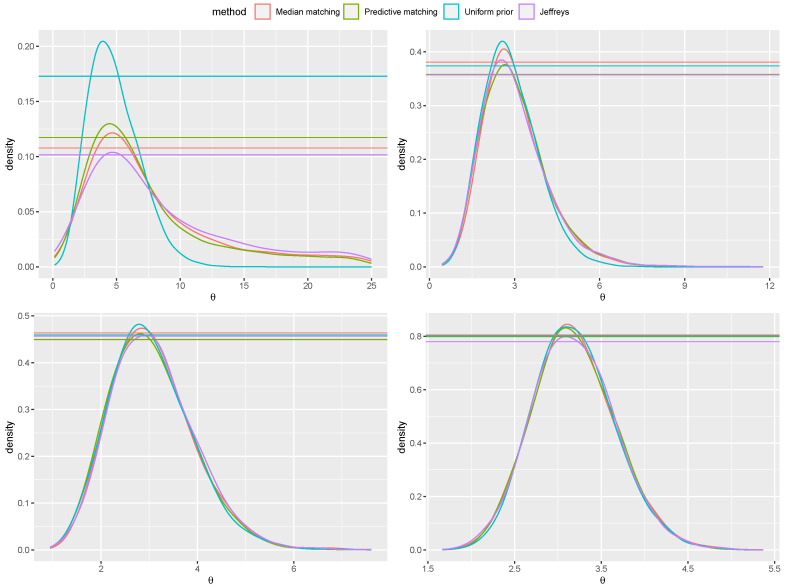
Inference for the scalar parameter θ of the skew-normal model with sample sizes n=20,30,50,200 (**top-left**, **top-right**, **bottom-left** and **bottom-right** panels, respectively). The red line is used for the posterior obtained from the median matching prior, the green one for the predictive matching prior, the violet one for the Jeffreys’ prior and the blue one from an improper flat prior. The horizontal lines identify the corresponding tangential sets associated to the hypothesis $H_{0}:\theta =3$.

**Figure 2 entropy-26-00058-f002:**
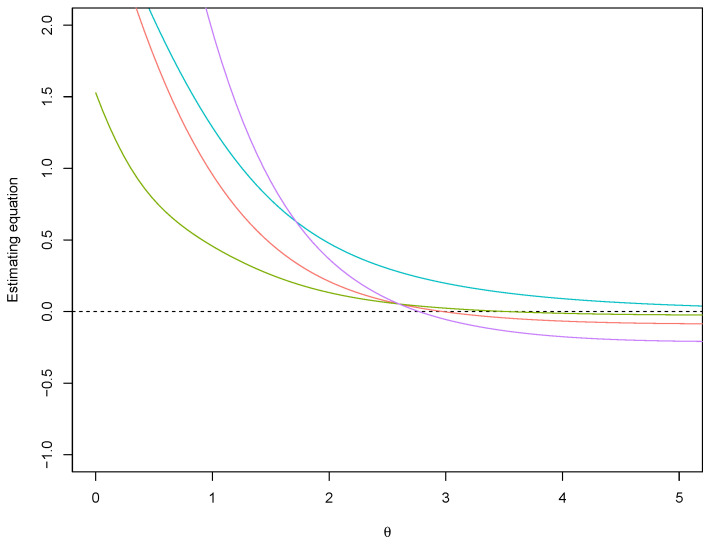
Skew-normal model: An example of ∂logπ(θ|y)/∂θ (estimating equation) with a flat prior (blue line), the median matching prior (red line) the predictive matching prior (green line) and the Jeffreys’ prior (violet line) in a sample where all the observations are positive.

**Figure 3 entropy-26-00058-f003:**
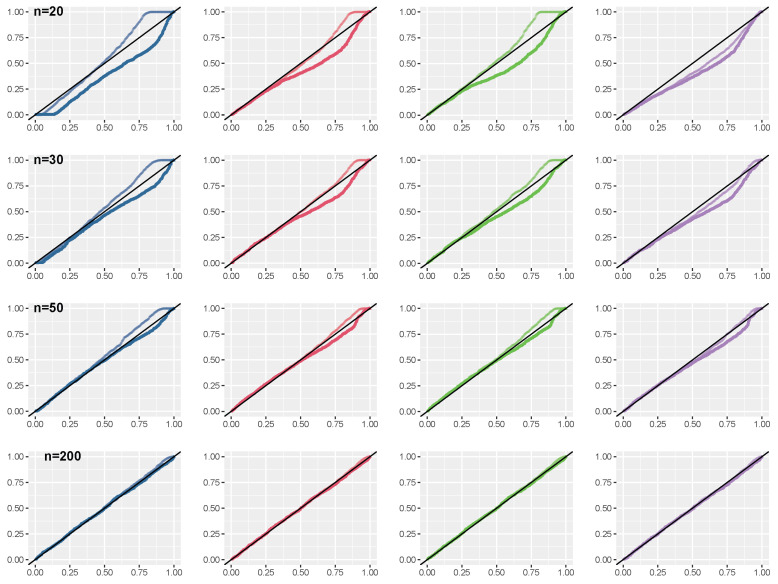
Skew-normal model: Distributions of e-values from a simulation study under the null hypothesis $H_{0}:\theta =3$, using a flat prior (blue line), the median matching prior (red line), the predictive matching prior (green line), and the Jeffreys’ prior (violet line). The darker line is used for the approximation (Equation 11), while the lighter is for that based on (Equation 12).

**Figure 4 entropy-26-00058-f004:**
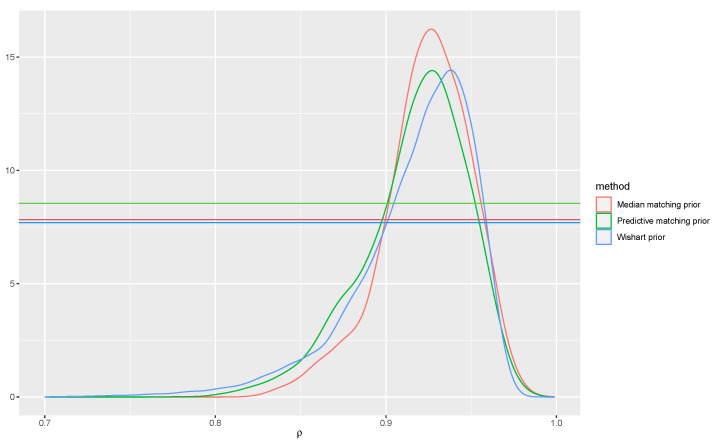
Posterior distributions for the correlation parameter ρ of the bivariate regression model obtained from MCMC draws and the three different priors.

**Figure 5 entropy-26-00058-f005:**
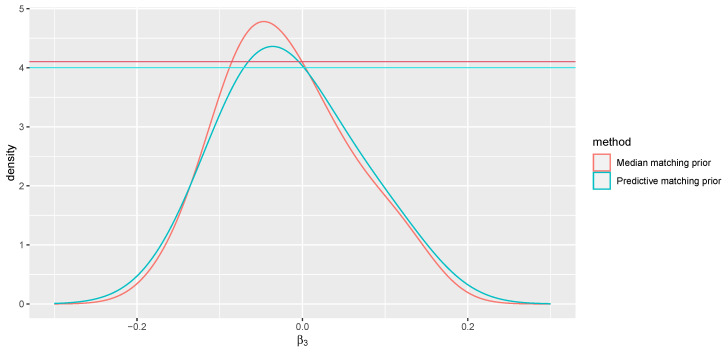
Median matching posterior distribution and predictive matching posterior distributions for $\beta_{3}$ in the logistic regression model.

**Table 1 entropy-26-00058-t001:** Skew-normal: e-values associated to the hypotheses $H_{0}:\theta =3$ and $H_{0}:\theta =4$.

Hypothesis $H_{0}$	*n*	Flat Prior	Median Matching Prior	Predictive Matching Prior	Jeffreys’ Prior
θ=3	20	0.59	0.62	0.56	0.65
	30	0.65	0.70	0.73	0.64
	50	0.81	0.84	0.82	0.91
	200	0.79	0.79	0.81	0.82
θ=4	20	0.82	0.91	0.98	0.84
	30	0.17	0.22	0.22	0.22
	50	0.20	0.20	0.20	0.21
	200	0.07	0.08	0.08	0.09

## Data Availability

Data and code used in the examples are available in https://github.com/elenabortolato/invariant_evalues (accessed on 28 November 2023).

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
