# Peer review of "Objective Priors for Invariant e-Values in the Presence of Nuisance Parameters"

_entropy, 2024, doi:10.3390/e26010058_

Round 1

Reviewer 1 Report

Comments and Suggestions for Authors

This paper develops asymptotic expansions of the posterior distribution in the context provided by the e-value and the FBST theory for testing sharp statistical hypotheses. 

These asymptotic expansions of the posterior distribution are especially useful in models containing nuisance parameters, for they allow a significant reduction in the computational cost for estimating the main quantity of interest, namely, the e-value ev(H|X). 

However, the methods developed in this paper are much more than fast-converging numerical approximations of the quantities of interest, even though the task of developing such fast-converging numerical approximations would be important enough. 

Nevertheless, methods developed in this paper also reveal interesting and important theoretic properties of pseudo-posterior distributions and matching prior distributions (specified by alternative matching goals like MAP, median, predicative targets, etc.)  

The Most noticeable theoretical property of the procedures derived in the paper is the invariance of the estimated e-values. 

The use of matching priors in the frequentist and Bayesian statistical literature has its own independent motivations. 

In particular, the use of matching ensures, up to the desired order of asymptotics, an agreement between Bayesian and frequentist estimates. 

The fact that matching priors, derived by such strong and intuitive "matching properties", end up ensuring "invariance properties" in the context of the FBST is remarkable! 

It reveals deep connections between the theoretic context of the FBST, and the theory of asymptotic expansions in statistics in general and that of matching priors in particular. 

The development of good asymptotic methods for fast and reliable numerical approximation would be enough to publish this paper. 

The numerical experiments presented in the paper clearly show that this goal is achieved in several models of interest. 

However, the already mentioned theoretical properties studied in this article also deserve further attention, study, and exploration.   

My main advice to the authors is to include a section of topics of further research, that should include at least two topics: 

(1a) Development of a library of computational routines exploring the methods developed in this paper for a wide range of statistical models of interest. 

(1b) Development of semi-automated procedures for further expanding this library. 

(2a) Further exploring the theoretical connections between e-value invariance and matching priors.  

(2b) Exploring the existence of similar connections in other classes of pseudo likelihoods.   

My congratulations to the authors for a paper that, on the one hand, is very practical and useful and, on the other hand, is beautiful and reveals deep theoretic properties in mathematical statistics.    

Reviewer 2 Report

Comments and Suggestions for Authors

I enjoyed reading this paper and learnt many new ideas from it. However, I was not familiar with other papers in this area and there seem to be so many approaches to the construction of pseudo likelihood functions that i was left a little confused about how this paper added or improved on the literature in specific ways. That said, I think the authors have done an excellent job in developing and presenting their ideas

Reviewer 3 Report

Comments and Suggestions for Authors

See attached report.

Round 2

Reviewer 3 Report

Comments and Suggestions for Authors

See my attached report.
